# A Novel Framework Based on Deep Learning Architecture for Continuous Human Activity Recognition with Inertial Sensors

**DOI:** 10.3390/s24072199

**Published:** 2024-03-29

**Authors:** Vladimiro Suglia, Lucia Palazzo, Vitoantonio Bevilacqua, Andrea Passantino, Gaetano Pagano, Giovanni D’Addio

**Affiliations:** 1Department of Electrical and Information Engineering (DEI), Polytechnic University of Bari, 70126 Bari, Italy; vladimiro.suglia@poliba.it (V.S.); lucia.palazzo@icsmaugeri.it (L.P.); vitoantonio.bevilacqua@poliba.it (V.B.); 2Scientific Clinical Institutes Maugeri SPA SB IRCCS, 70124 Bari, Italy; andrea.passantino@icsmaugeri.it (A.P.); gianni.daddio@icsmaugeri.it (G.D.); 3Apulian Bioengineering S.R.L.,Via delle Violette 14, 70026 Modugno, Italy

**Keywords:** human activity recognition, activities of daily living, inertial measurement units, time-series, artificial intelligence, deep learning, convolutional neural networks, data augmentation, motion analysis, rehabilitation, bioengineering

## Abstract

Frameworks for human activity recognition (HAR) can be applied in the clinical environment for monitoring patients’ motor and functional abilities either remotely or within a rehabilitation program. Deep Learning (DL) models can be exploited to perform HAR by means of raw data, thus avoiding time-demanding feature engineering operations. Most works targeting HAR with DL-based architectures have tested the workflow performance on data related to a separate execution of the tasks. Hence, a paucity in the literature has been found with regard to frameworks aimed at recognizing continuously executed motor actions. In this article, the authors present the design, development, and testing of a DL-based workflow targeting continuous human activity recognition (CHAR). The model was trained on the data recorded from ten healthy subjects and tested on eight different subjects. Despite the limited sample size, the authors claim the capability of the proposed framework to accurately classify motor actions within a feasible time, thus making it potentially useful in a clinical scenario.

## 1. Introduction

The recent advances in medicine have improved life conditions and increased life expectancy so that healthcare systems have to cope with the aging of the global population [1,2]. In addition, there are multiple categories of people experiencing motor disorders, from Parkinson’s patients [3] to post-stroke individuals [4]. The psycho-motor frailty of these subjects can result in sedentary lifestyle choices that may aggravate their condition, thus raising the impact on the health system [5]. On the contrary, to pursue safety and well-being, the degeneration of their motor skills ought to be prevented by stimulating beneficial motor behaviors like an active lifestyle; therefore, recognizing activities of daily living (ADLs) can help monitor human habits and assess motor actions [6].

The scientific literature has given more and more attention to the field of human activity recognition (HAR), which aims to classify human actions by exploiting sensor data [7]. HAR has covered various contexts, from industry [8,9] to sport [10], but a wider application lies in the medical field [7,10,11,12,13,14]: in this realm, subjects’ activities can be remotely registered outside the clinic [15] and clinicians can evaluate their functional abilities after treatment [16,17]. HAR can also enhance a rehabilitative program inside the clinic for the sake of an assist-as-needed approach: in particular, recognizing the motor actions performed by patients (e.g., post-stroke individuals or people with psychomotor dysfunction) can allow for correcting motions or encouraging further exercise when required [18].

In addition, even patients with mental disorders (e.g., children with autism spectrum disorder) can be continuously monitored so that stereotypical actions (e.g., arm flapping) that are symptoms of anxiety may be identified and promptly counteracted [19].

A typical HAR experimental protocol encompasses a set of activities that the subject is asked to perform. These motor tasks may involve mainly upper body [9,20], or lower body (e.g., walking or climbing/descending stairs) [15,17,21,22], or even require the individual to drive upper- and lower-extremities in a proper combination (e.g., lying in bed) [23,24,25,26,27,28,29]. Furthermore, the protocol to collect data for HAR purposes tends to be designed such that ADLs are performed separately, i.e., a batch of repetitions of one of the activities to be recognized is asked to be executed by every subject [30]. However, this separate execution does not account for the continuous nature of human activities, which are more likely made successively [31]; therefore, the data collection should entail the recognition of uninterruptedly performed ADLs, namely continuous human activity recognition (CHAR) [32]. Such workflow can address the natural transition from one activity to another that humans execute in daily life, thus making the recognition system more spendable in the field of remote health monitoring [31].

A framework targeting HAR comprises two main components: the acquisition system, which collects several signals that are descriptive of the movement performed by the subject, and the classification pipeline, which processes the collected data and returns the type of activity [29].

The acquisition system may differ in the type of adopted sensors and the modality employed to acquire the signal describing human movement. There exist two main categories in which to classify the type of sensors: fixed sensors (e.g., videocameras, proximity and light sensors) are installed at specific locations of the environment and monitor activities in a confined area, whereas wearable sensors are directly worn by the subject, as in the case of inertial measurement units (IMUs), pressure and heart rate sensors [5]. Though being previously utilized to accurately label activities [6,15], fixed sensors like cameras are not very suitable when ADLs execution requires subjects to move outside the area covered by them [24]; besides, cameras suffer from variable illumination, occlusion occurrence, presence of shadows, and time-varying contrast, especially in outdoor environments; such disadvantages, together with privacy issues and their lack of portability prevent them from continuously monitoring human activities [33]. In light of these limitations, most of the research in the HAR field, especially for remote monitoring, has preferably adopted wearable sensors because of their low cost and higher flexibility in providing continuous monitoring [34].

The number of types of sensors comprised in a HAR-oriented setup leads to distinguishing two approaches for data acquisition, which are unimodal and multimodal approaches. The unimodal approach refers to the use of only one modality (i.e., type of sensor) [8,26,27,28,35], whereas the multimodal approach aims to integrate data from different sources by using multiple types of sensors, e.g., wearables like electromyography (EMG) sensors [22]. However, EMG signals are not widely used in HAR frameworks since their measurement are affected by electrical noise and motion artifacts due to human sweat [36]. Therefore, most of the frameworks addressing HAR have focused on the unimodal approach, thanks to which information from different sensors of the same type (e.g., IMU sensors) can be integrated [2,37,38,39].

Ultimately, the acquisition paradigm of a HAR system exploiting IMUs may change according to the configuration of the sensors, which is given by both their number and their location on the human body. The amount of sensors in a HAR-targeting setup depends on the activities to be recognized: the exploitation of a single sensor may be enough when ADLs require to drive only one degree of freedom of one human joint (e.g., wrist flexion/extension in case of upper-limb driven actions [20] or leg extension from sitting position and leg flexion from standing position for lower-limb tasks [40]); on the other hand, a higher number of sensors is needed in case of more complex ADLs that target multiple joints (e.g., walking, sitting on a chair, lying-down on a surface). The activities to be classified have an impact on the sensor placement, which may be determined starting from anatomical landmarks which are body areas that are close to anatomical points of interest (e.g., lower-limb joints for recording human locomotion).

The recognition of human actions in a workflow targeting HAR requires a pipeline entailing a sequence of steps that may include data processing, feature extraction and artificial intelligence (AI) techniques to perform classification: at first, the signals acquired from sensors are processed to reduce noise [41], cope with missing values and remove possible artifacts [9,42]; secondly, data are segmented to identify the portion of the preprocessed signals that are informative of the executed activities [43]; signals can be optionally converted into images as well [15,17,20,28,44,45]; afterward, features are extracted for each segment from either images or time-series data [1,2,38,46] to capture meaningful characteristics of the performed activities; ultimately, these features and their corresponding ground truth labels are used as input to train a classifier, whose performance is evaluated based on quantitative criteria, such as accuracy [47].

The HAR-oriented pipeline may differ according to the AI model used to discriminate ADLs. On the one hand, Machine Learning (ML) procedures are trained on hand-crafted features [24,29], but implies a manual extraction based on domain knowledge that can be increasingly time-consuming as the dataset dimensionality enlarges due to the need for the high amount of repetitions and subjects for the sake of generalizability [28]. On the other side, Deep Learning (DL) architectures can be directly fed by raw data and automatically learn patterns through the process of backpropagation without any prior knowledge of the signals [47].

Convolutional Neural Networks (CNNs) are the most widely employed among the DL architectures proposed in the studies addressing HAR [8,22,23,27,32,38,43,46]. CNNs usually work on images by means of two-dimensional convolutions for practical problems as defect detection [47,48]; notwithstanding, their one-dimensional variant is preferred because it allows working directly on time-series signals instead of their corresponding images, thus reducing the computational cost [32]. Furthermore, CNNs employed in HAR frameworks may have either a sequential or a multi-branch structure: in the former case, layers process all the IMU signals of the input dataset [23], whereas in the latter case each branch, which may be fed by one of the IMUs included in the experimental setup, is computed in parallel with the others [8,27].

The majority of DL-based HAR-oriented workflows test classifiers on inertial data that are related to a separate execution of ADLs. Only a few works test the DL architecture on data that come from an uninterrupted sequence of activities and are previously trained on IMU signals corresponding to stand-alone activities [30,31,32,49,50]. Most related works proposed a CHAR-oriented approach with a setup based only on radar sensors [30,31,49,50]; however, their applicability to outdoor environments is limited by their measurement area [30]. Furthermore, to the best of our knowledge, there is only one work that performs CHAR with DL methods based only on kinematic data: Jaramillo et al. measured the evolution of the hip joint angle by means of IMU sensors and encoders integrated in an exoskeleton, and included both separate and continuous acquisition protocols [32].

In view of the above-mentioned works, a paucity in the literature has been noticed regarding the implementation of a DL algorithm that is fed by inertial measures and is capable of recognizing ADLs executed in a continuous way. Hence, the goal of this article is to present the design, development, and test of a framework that aims to perform CHAR by training a custom 1D-CNN with IMU signals that are related to a separate execution of ADLs and testing it on inertial data recorded during an uninterrupted execution of those activities. For this purpose, one experimental protocol with interrupted execution of tasks by ten healthy subjects is used to train the DL-based architecture, whereas another protocol involving a seamless execution of those motor actions is employed to test the CHAR workflow on eight healthy individuals (different from the first ten ones). Further, a classification strategy is proposed to enhance accuracy without increasing significantly the time needed to recognize the activity. The model performance is evaluated for different combinations of the adopted sensors in order to determine the optimal configuration.

This paper is organized as follows: Section 2 describes materials, which includes the system for collecting inertial data during the execution of ADLs, and the methodology adopted in this work, which comprises a preprocessing phase, a DL architecture based on a custom CNN, and statistical analysis. The outcomes of the ADLs classification are provided and discussed in Section 3 and discussed in Section 4. Ultimately, Section 5 draws the final remarks about the conducted study and delineates ideas for future works.

## 2. Materials and Methods

The framework that is proposed to address continuous human activity recognition is comprised of two main stages, which are a data collection stage explained in Section 2.1, and a classification pipeline reported in Section 2.2.

The acquisition of inertial data is accomplished by means of four IMU sensors, whose components are given in input to a DL-based model. For the sake of performing CHAR, this architecture is trained on IMU signals coming from multiple separate executions of four ADLs before being tested on the inertial data that are related to a multiple uninterrupted execution of motor tasks. This framework targeting CHAR is depicted in Figure 1.

### 2.1. Data Collection

#### 2.1.1. Participants

Eighteen healthy subjects (34.94±11.58 years old, eight males) are recruited from the staff of the IRCCS Maugeri (Bari, Italy). These participants differ in age, weight, height, and anthropometric characteristics (e.g., length of body segments) for the sake of higher data heterogeneity.

All subjects are right-handed with no motor or cognitive pathologies. Each subject is informed about the execution of the required activities prior to the experimental session. Besides, they were asked to perform tasks freely (i.e., with no restrictions on their body movements) to resemble a daily life situation.

#### 2.1.2. Activities

The activities to be executed were defined in accordance with the clinical staff of the IRCCS Maugeri. Four ADLs were selected among the ones identified as the most frequently performed in everyday life in a survey about ADL occurrence in HAR datasets [51].

The ADLs of this study differ in their biomechanical characteristics and can be described as follows.
-*Walking* on a surface with no asperities mainly requires alternating flexion-extension movements of the three lower limb joints (i.e., hip, knee, ankle) and may be combined with arm swinging. Note, that leg motion during walking is often associated with arm swinging to provide increased balance.-*Turning* while walking typically requires the coordination of various movements, including the intra- and extra-rotation of both lower-limb joints (mainly hips) and trunk.-*Sit-to-stand* transition, i.e., rising from a chair, principally entails hip and knee extension, as well as trunk rotation to bend/straighten the torso for keeping balance; it possibly involves additional leverage on arms or hands when the individual needs them for a lift that is both greater and safer.-*Lying-down* on a surface (e.g., a couch), consists of two phases: the subject first reaches a sitting position, and then moves to the lying position through motor actions that mostly include hip and knee flexion/extension, trunk rotation and hip and knee abduction/adduction. In addition, this transition from the standing to the lying position can be supported by recruiting the wrist to lean hands on the couch for the sake of either a safer or a more comfortable motion. Subsequently, this motor action is completed by fully relaxing the body with the flexion/extension of the hip, knee, and trunk, as well as by resting their hands and arms on the bed.

#### 2.1.3. IMU Sensors

The number and placement of IMUs must be properly designed since they can have an impact on the performance of a HAR pipeline.

Hence, four sensors have been chosen and placed on the two sides of the human pelvis, the right wrist (i.e., the wrist of the dominant arm), and the sternum (see Figure 2), since they are anatomically close to the human joints that are driven during the execution of the selected ADLs. More in detail, the bilateral placement on the pelvis enables accurate monitoring of the pelvis movements; the sensor on the right wrist is essential for recording the arm swing that assists walking, as well as the use of hands to support lying-down action; the sensor placed on the sternum is useful to monitor the trunk, which mainly contributes in sit-to-stand and lying-down activities.

These sensors are attached to the subjects by straps because they are easy to wear and adaptable to the different body sizes of the subjects.

The experimental data are collected with the Motion Studio system by APDM (APDM Inc, Portland, OR, USA, https://apdm.com (accessed on 15 January 2024)). The system (see Figure 3) consists of the following three main components:-a set of wireless body-worn IMUs, called OpalTM sensors, measuring 43.7 mm × 39.7 mm × 13.7 mm (L × W × H), each with a docking station;-an Access Point for wireless data transmission and synchronization of the independent sensors;-the Motion Studio software to manage the acquisitions of the recorded data.

Each Opal sensor is wireless connected by Bluetooth communication protocol to a remote PC, and includes a 3-axis 14-bit accelerometer to measure linear acceleration, a 3-axis 16-bit gyroscope to record angular velocity, and a 3-axis 16-bit magnetometer for magnetic field intensity [52].

The Motion Studio software was used to record data in real time with a sampling rate of 128 Hz. Each recording session returns signals from the accelerometer, gyroscope, and magnetometer, whose combination has already proven to outperform a subset of IMU components in related works about HAR [53].

#### 2.1.4. Experimental Protocol

The experimental sessions are performed in the MARLab—Movement Analysis and Robotic Lab—of the IRCCS Maugeri in Bari (Italy). The protocol took approximately 40 min to be completed by each subject. The authors have proposed a twofold experimental protocol in order to entail the separate execution of ADLs for training and seamless ADLs for testing the proposed model. As a result, two datasets have been acquired to accomplish CHAR with the DL-based classifier.

One dataset, which is used for training the model, used 10 subjects (four males and six females) who performed each activity multiple times and separately, i.e., each subject repeated the three previously defined tasks one by one and repeated each of them 10 times. This dataset contains a total of 300 acquisitions, each corresponding to a specific activity. The experimental protocol for the training dataset has included three tasks (see Figure 4a) that are aimed at acquiring data related to the chosen ADLs. These motor actions have been conducted as follows.
-*Walking+Turning task*: the subject stands quietly for 30 s, walks for 7 m, turns 180 degrees counterclockwise around a pin, and walks back to the start point. In the end, the subject has to stand quietly for 5 s [52].-*Sit-to-Stand task*: the subject sits on a chair with their heels at a reasonable distance for a comfortable execution of the task and keeps this position for 5 s. Next, the subject rises from the chair to reach a standing position, which they keep for another 5 s.-*Lying-down task*: the subject keeps standing with heels at a reasonable distance from the bed for a comfortable execution of the task and keeps this pose for 5 s. After that, the subject lies down such that he/she feels comfortable, and keeps this lying position for 5 s.

A reasonable waiting time was allowed between two subsequent repetitions of each activity to prevent the subject’s fatigue, which can alter the results [54,55].

On the other hand, the other dataset, which is employed for testing the model used eight subjects (four males) who executed the above-mentioned activities continuously in a specific order, i.e., each subject performed tasks with no interruption in a predefined circuitry resembling a daily life scenario. Therefore, the experimental protocol for the test dataset is made up of the same ADLs as those of the training dataset, but such motor actions have been conducted in order to follow a predefined circuit (see Figure 4b): each participant starts from a sitting position and stands quite for 5 s; next, he/she gets up from the sitting position, walks for 7 m, turns clockwise 180 degrees, and then walks back for 7 m to reach the couch; after that, he/she turns in the preferred direction, lies down and keeps lying for about 5 s. After completing one execution of the circuit, subjects were instructed to wait a fair amount of time before the next repetition to prevent fatigue [54,55]. Each recording of the test dataset is repeated five times per subject, for a total of 40 acquisitions.

### 2.2. Classification Pipeline

This Section elucidates the CHAR-addressing pipeline, which includes a preprocessing stage (Section 2.2.1), the details about the architecture and the training of the custom CNN classifying ADLs (Section 2.2.2). Furthermore, the strategy for achieving the final prediction of activities is described in Section 2.2.3, whereas the metrics used to evaluate the classification performance for all sensor combinations are reported in Section 2.2.4 and compared in Section 2.2.5.

#### 2.2.1. Preprocessing

##### Normalization

The physical variable measured by the IMUs (see Section 2.1.3) makes the values of one IMU component stay within a different range with respect to the values of the other components. This may result in extreme differences among input data and can worsen the capability of recognizing ADLs, as it is more complex to detect patterns in the data [56]. Therefore, a normalization operation is needed to achieve a uniform representation of the data. More specifically, the data of each IMU component have been normalized to lie in the range [−1,1], as conducted in other related works about CHAR [31].

##### Segmentation

Signals coming from the IMUs used in a HAR workflow may comprise data that are not related to the motor tasks of interest, especially when some static periods (e.g., keeping a standing/standing/lying position) or transitions between two ADLs (e.g., stand-to-walk and walk-to-stand) are included in the dataset. Therefore, a data segmentation phase is needed to identify the time frames in which data streams might contain relevant information about the target ADLs. Indeed, data were manually segmented by means of a signal inspection both for training and test datasets. These segments are then associated with a label representing the ground truth for the recorded activity for the sake of a supervised learning strategy [47].

##### Windowing

A windowing procedure is applied both for training and test datasets to obtain an even higher amount of data to feed the proposed CHAR model. The window width must be informative enough to capture the performed activity [17]; nonetheless, an excessively wide window must be avoided to prevent a high computational cost and a classification delay that is not apt for the specific application [43]. As a consequence, it has been decided to adopt a step size of 128 samples, corresponding to 1 s, and an overlap of 64 samples, corresponding to 0.5 s. Such windows may be part of a longer window for which to classify ADLs [57].

##### Resampling

The duration of one activity execution may change depending on the subject’s characteristics, considering that an individual with motor disorders needs more time than a healthy one to accomplish the same task [58,59,60]; this results in a different number of windows that could not cover the entire length of the signal, thus causing a loss of information for the network. Therefore, IMU data are resampled such that the new signal length is an integer multiple of the window width to ensure that the windowing procedure keeps the whole signal.

##### Data Augmentation

Several factors can influence inertial data recorded for a CHAR experiment, thus making the ones related to a continuous protocol—i.e., related to a seamless execution of human activities—differ from those of a discontinuous protocol, i.e., related to a separate execution of ADLs. For instance, the structural characteristics of the environment may lead to rearranging the starting point of one motor action in order to keep the continuous nature of the protocol. As an instance, the starting point may swap with the ending point, thus changing the motion direction of the path related to the continuous protocol with respect to that of the discontinuous protocol, i.e., the subject turns counterclockwise in the former, but clockwise in the latter one. This gap in the operation condition is reflected in some components of the inertial signals from which the DL-based model learns patterns, thus potentially worsening the performance of the activity recognition [47].

Hence, for each subject of the training dataset, signals have been treated with a conventional data augmentation technique by flipping to reproduce the operational condition of the test dataset and improve the classifier robustness [61]. More specifically, Figure 5 visually reports the normalized magnetometer signal of the IMU sensor located at the left hip during the execution of sit-to-stand and turning tasks related to the test dataset (i.e., continuous execution), and to the training dataset (i.e., separated execution) before and after data augmentation. The operational discrepancy leads the magnetometer signal to be flipped along the x-axis and z-axis for the turning action and along the x-axis and y-axis for the sit-to-stand task. Hence, such components of the training dataset have been flipped to make the magnetometer components of the training dataset resemble those of the test dataset.

On the other hand, the operational discrepancy is not reflected in accelerometer and gyroscope signals because, differently from the magnetometer data, linear acceleration, and angular velocity do not change according to the subject’s position with respect to the position of the magnetic north. Hence, such components of the IMU sensors are not affected by the change in motion direction and they have not been flipped.

In addition, the time needed to perform ADLs can be different for intrinsic characteristics of the task included in the protocol, since motor actions such as walking for seven meters last far more than standing from a chair. This difference in activity duration results in a different number of samples (i.e., windows) that feed the classifier; consequently, the dataset of IMU signals would be unbalanced towards the majority class (i.e., the motor action with the highest number of windows), thus leading to reduced classification performance [47]. Therefore, another conventional data augmentation technique for time series data is applied to compensate for the imbalance [61]. In particular, the signals of the minority classes (e.g., sit-to-stand, turning, lying-down) are scaled by a factor that can be either amplified or attenuated so as to simulate slight magnitude differences among the repetitions made by one subject. The number of trials of the above-mentioned activities has been increased, thus enlarging the volume of the collected data.

##### Sensor Combinations

Sensor configuration, which is given by the number and placement of the IMUs adopted in the setup, can have a significant impact on the performance of the HAR system [39,46,56,62]. Therefore, it might be useful to evaluate the different combinations of the sensors from which the inertial data feeding the model come. More specifically, the authors focused on those combinations involving two and three sensors, as well as all the four IMUs explained in Section 2.1.3.

In light of this, each window of the pre-processed dataset is an array of size $W_{l}\times N_{c}$, where $W_{l}$ is the window width and $N_{c}$ is the number of sensor channels, which differs according to the combination of sensors to be evaluated: it is six for any sensor pair, nine for any sensor triple, and twelve for the combination with four sensors.

#### 2.2.2. Custom Convolutional Neural Network

In this work, the authors have employed a one-dimensional CNN for classifying ADLs, since they were successfully applied in related works about HAR [8,22,23,27,43,46] and CHAR [32]. Specifically, the chosen architecture (see Figure 6) employs three distinct parallel branches to process signals from the accelerometer, gyroscope, and magnetometer simultaneously. This structure automatically extracts features from signals that have different physical meanings, thus allowing it to leverage all the data recorded by each sensor at the same time.

The input layer is fed by a multidimensional array whose shape is $\left(N_{w},W_{l},N_{c}\right)$, where $N_{w}$ is the number of windows in the input dataset, which may change across subjects and trials, $W_{l}$ is the window length, which is fixed, and $N_{c}$ is the number of sensor channels, which differs according to the combination of sensors to be evaluated (e.g., it is six for any sensor pair, nine for any sensor triple, and twelve for the combination with four sensors).

A grid-search method is employed to optimize the architectural characteristic of the 1D-CNN, i.e., determine the number of convolutional and dense layers, as well as the number of neurons that maximize validation accuracy [9,38]. Therefore, each branch consists of two 1D convolutional layers using 128 filters and kernels of size 5 for the first and 3 for the second one, and one max-pooling layer. Next, two 1D convolutional layers use 64 filters and kernels of size 5 and 3, respectively, followed by a max-pooling layer. Then, a flattened layer reshapes data into a linear vector. All convolution layers are characterized by a *ReLu* nonlinear activation functions. Subsequently, all three branch outputs are concatenated in one linear vector, thus gathering the smaller previous outputs. This composite feature vector is then fed through two fully connected layers, each with 128 neurons, for learning more abstract representations of the data. The first one is followed by a dropout layer that switches off 20% of the neurons to prevent overfitting. Finally, an output layer with a softmax activation function is included to classify tasks into one of the four defined classes. This layer is responsible for the final prediction of activity labels, based on the representations learned from the input data.

The same heuristic on validation accuracy is used to select the model hyper-parameters, such as the optimizer, the learning rate, the batch size, and the number of epochs. The best performance is obtained by using a number of epochs of 200, a batch size of 64, and the Adam optimizer with a learning rate of 0.001. An early stop criterion monitoring the loss value on the validation set during the training is exploited; however, criteria accounting for other metrics (e.g., validation recall) can be used. Consequently, the value of the patience is set to 10 to stop the learning process prematurely if the value increases for 10 consecutive iterations on the validation batches.

For each combination of sensors, the dataset of the separate execution of ADLs was split into ten stratified folds, 75% of which is assigned for training and the remaining 25% for validation. Afterward, a 10-fold-cross-validation methodology is employed to ensure a fair and unbiased evaluation of the model [48]. All investigations in this study are conducted on the Google Colab-Pro framework to train the model on a Tesla T4. Tensorflow, Sklearn, Pandas, and Numpy libraries have been exploited for training and inferencing the CHAR-targeting architecture.

#### 2.2.3. Classification Strategy

Human activities are usually classified by applying a sliding-window technique on the input data, meaning that the prediction is given for each of the windows in which the signal is divided [17]. However, signal duration within a trial can differ across subjects, because people with motor impairment may need a higher amount of time to accomplish the task [58,59,60]. In light of this, monitoring ADLs for pathological individuals could admit a slightly slower classification by means of wider windows to enhance the accuracy of the final prediction of the performed motor action.

Hence, for each activity, the authors propose a classification strategy that entails the combination of predictions coming from *sub-windows* of a single trial to achieve the model prediction related to a *grouped window* that lasts as in the trial [57]. Ultimately, the classification of the *grouped window* is given by the average of the predictions made on the *sub-windows*.

#### 2.2.4. Performance Metrics

The efficacy of the CHAR-oriented framework has been evaluated by means of two metrics. On the one hand, classification performance is measured through accuracy, since the input dataset has been rebalanced by means of data augmentation. The formula of this performance index is given below: (1)$$\mathrm{ACC}=\frac{TP+TN}{TP+TN+FP+FN}$$
In such equations, TP, TN, FP, and FN represent true positives, true negatives, false positives, and false negatives, respectively.

On the other hand, the feasibility of the framework in a real-time clinical application is investigated by computing inference time [32]. For each ADL, this metric is calculated both for the *sub-windows* and *grouped windows* of each trial as follows:-*sub-windows* inference time is the time that is necessary for returning the prediction from a single sub-window;-*grouped-window* inference time is the time needed for returning the prediction from the single trial of the activity.

Hence, this metric is mathematically defined as follows:(2)$$\;\mathrm{IT}\;\left[s\right]=t_{end}-t_{start}$$

In such equations $t_{start}$ is when the inference procedure is started by giving the data in input, whilst $t_{end}$ is when the inference process differs depending on the window type. More specifically, given a single trial of each ADL, $t_{end}$ may be:-the time in which the model returns a classification label for a single *sub-window*, in the case of *sub-windows* inference time;-the time in which the model gives the classification output as the average prediction across *sub-windows* in the case of *grouped-window* inference time.

The proposed metrics are computed for each test fold to obtain a confidence interval distribution for each index [48].

#### 2.2.5. Comparisons and Statistics

Sensor placement can be impactful on the model performance of a workflow for recognizing human activities [39,46,56,62]. Hence, the model has been tested on all the combinations entailing at least two of them, which are detailed in the following.
Combinations with two sensors:
-*Right Pelvis + Left Pelvis* (RP+LP);-*Left Pelvis + Sternum* (LP+S);-*Right Wrist + Left Pelvis* (RW+LP);-*Right Pelvis + Sternum* (RP+S);-*Right Wrist + Right Pelvis* (RW+RP);-*Right Wrist + Sternum* (RW+S);Combinations with three sensors:
-*Right Pelvis + Left Pelvis + Sternum* (RP+LP+S);-*Right Wrist + Right Pelvis + Left Pelvis* (RW+RP+LP);-*Right Wrist + Left Pelvis + Sternum* (RW+LP+S);-*Right Wrist + Right Pelvis + Sternum* (RW+RP+S);Combination with all the four sensors:
-*Right Wrist + Right Pelvis + Left Pelvis + Sternum* (RW+RP+LP+S).
Such combinations have been statistically compared with the non-parametric Friedman’s test since the hypothesis of a Gaussian distribution is excluded due to the limited number of tests. Besides, a pairwise post-hoc test with Bonferroni’s correction was performed with a significance level set to *p* < 0.05. These analyses were conducted using MATLAB 2022b.

## 3. Results

This section presents the results of the continuous human activity recognition performed with the proposed DL-based framework: the outcomes of the efficacy of the classification strategy are reported in Section 3.1, whereas the results concerning the investigation of the optimal sensor configuration are described in Section 3.2.

### 3.1. Differences between Window Types

The classification performance of the CNN-based framework addressing CHAR has been evaluated in each fold for all combinations of sensors. Hence, many distributions have been obtained for the two window types described in Section 2.2.3, i.e., sub-windows of a single repetition of the ADL and *grouped window* lasting the repetition itself, and for all metrics mentioned in Section 2.2.4, which are accuracy and inference time.

Since two classification strategies have been applied, i.e., without and with averaging, the authors have compared the distributions of accuracy and inference time to investigate the impact of the averaging technique on both these metrics.

Table 1 contains the average accuracy of the CHAR-oriented model computed on ten-fold testing sets for each combination of sensors before and after averaging predictions.

The accuracy of the DL-based classification has been significantly boosted for almost all combinations of sensors passing from sub-windows to a grouped window, with an increment of about 15% for RW+LP and RW+LP+S with p<0.001; however, the model tested with data related to RW+RP+LP outperforms the outcome corresponding to any other combination for both window types, since the classifier accuracy has significantly raised to 96.69% with p<0.001.

Moreover, the average and standard deviation of inference time for 10 testing sets of each combination of sensors for each window type are reported in Table 2.

The proposed 1D-CNN needs almost 300 ms on average with a low standard deviation to recognize one activity performed with no interruption with the other ones. Furthermore, the average inference time has incremented, but not significantly (p>0.05), for all combinations of sensors. Notwithstanding, such time stays in the order of magnitude of a few milliseconds.

### 3.2. Differences in Accuracy among Sensor Combinations

Statistically significant differences were revealed for each performance index when comparing sensor combinations. The outcomes of the proposed metric for evaluating the efficacy of the proposed framework to address CHAR are pictorially depicted in the boxplot reported in Figure 7.

Friedman’s test revealed that ACC significantly differs among sensor combinations, with p<0.01.

Regarding the comparisons among sensor configurations with two sensors, according to post-hoc tests, the ACC in the RW+LP combination is significantly greater than the ACC in both LP+S (p<0.01) and RP+S combinations with p<0.05, as well as than the ACC in RW+RP combination with p<0.01. Instead, no statistically significant differences were found in the values of ACC between the RW+LP combinations and any other combination with two sensors. However, the RW+LP combination was revealed to be the best configuration with two IMUs placed at the *Right Wrist* (RW) and the *Left Pelvis* (LP). Similarly, slightly worse performance can be observed for the RP+LP and the RW+S combinations.

Concerning the comparisons between each configuration with two sensors and each one with three sensors, posthoc tests showed that ACC in the RP+LP+S combination revealed no statistically significant difference with any sensor pair On the other hand, ACC in the RW+RP+LP combination is significantly higher than ACC in the LP+S combination with p<0.001, as well as ACC in RP+S and RW+RP combinations with p<0.01. Secondly, ACC in the RW+LP+S proved to be better than ACC in both RP+S with p<0.01, as well as higher than ACC in LP+S and RW+RP with p<0.001. Besides, ACC in the RW+RP+S combination significantly lessens ACC in the RW+LP combination with p<0.05, but no other significant differences were found with any other sensor pair.

As for the comparisons among sensor triples, posthoc tests revealed that ACC in the RW+RP+S combination is the lowest one among sensor triples. More in detail, it significantly lessens ACC in the RW+RP+LP and RW+LP+S combinations with p<0.01; ACC in the RW+RP+S combination is also slightly worse, though not significantly, than ACC in the RP+LP+S triple.

Considering the comparisons between the sensor quadruple and any sensor triple, according to the post-hoc test, the ACC median in the RW+RP+LP+S combination is almost comparable with ACC in the RW+RP+LP and RW+LP+S triples; on the other side, ACC in the sensor quadruple is higher than the one in RP+LP+S, though with no statistically differences; in addition, the RW+RP+LP+S combination outperforms the RW+RP+S one in terms of accuracy with p<0.01.

Besides, this sensor quadruple is better than both LP+S and RP+S combinations with p<0.001 and p<0.01, respectively, as well as RW+RP with p<0.001. The ACC median in the RW+RP+LP+S combination is even higher, but not significantly, than the one in any other sensor pair.

The outcomes depicted in the boxplots can be further investigated by means of the confusion matrices of the ten-fold testing sets. More in detail, the confusion matrices of the predicted activities against ground truth using RW+LP+S, LP+S, RW+S, RW+RP+LP, RP+LP, RW+RP, RW+RP+LP+S, RW+RP+S, and RP+LP+S combinations are shown in Figure 8.

Regarding the confusion matrix related to the LP+S combination, a misclassification can be observed between sit-to-stand (S2S) and lying-down (LD): besides, walking (W) is confused with both turning (TN) and S2S. Similarly, the confusion matrix related to the RW+S configuration shows that S2S is still confused with LD, and W is misclassified with TN in an even worse way. On the contrary, The confusion matrix related to the RW+LP+S combination decreases such misclassifications and improves the overall accuracy.

The confusion matrix related to the RP+LP combination reveals a misclassification between S2S and LD, as well as walking W is confused with either turning TN or S2S. Similarly, the confusion matrix related to the RW+RP combination shows that this configuration leads to confusing S2S with LD, as well as worsening the misclassification between W and TN. On the other side, The confusion matrix related to the RW+RP+LP triple reduces the number of false positives and false negatives, thus leading to enhanced overall accuracy.

As for the confusion matrix of the RW+RP+S triple, S2S is confused with LD, whilst TN is misclassified with W; furthermore, the confusion matrix of the RP+LP+S combination reveals a comparable misclassification between S2S and LD, whereas W is slightly confused only with S2S. Instead, according to the confusion matrix related to the RW+RP+LP+S configuration, a decrease in false positives and negatives is produced when using all sensors, thus ensuring a better performance in terms of accuracy.

### 3.3. Differences in Inference Time among Sensor Combinations

In this subsection, the authors present the results in terms of inference time compared among sensor combinations. Such comparisons are pictorially depicted in the boxplots reported in Figure 9.

Friedman’s test leads to statistically significant differences in IT among sensor combinations. Besides, post-hoc tests revealed that IT in the RP+S combination is significantly lower than IT in both RW+LP+S and RW+RP+S triples with p<0.05, as well as inferior than IT in the RW+RP+LP+S configuration with p<0.01. Inference time stays below 300 ms in almost all sensor combinations, except for IT in the RW+RP+LP+S combination whose distribution reached the highest values (almost 450 ms), but less than 1 s.

## 4. Discussion

In this work, the authors present a DL-based framework that is aimed to perform continuous human activity recognition (CHAR), i.e., the classification of activities of daily living (ADLs) with a custom convolutional neural network (CNN) that is fed by data acquired by means of inertial measurement units (IMUs) located at four body parts, which are the left pelvis (LP), right pelvis (RP), sternum (S), and right wrist (RW). The experimental protocol requires the subject to perform four ADLs, which are *Walking* (W), *Turning* (T) while walking, *Sit-to-stand* (S2S) and *Lying-down* (LD) on a surface, in two ways: On the one hand, ADLs are executed separately (i.e., every subject performs multiple repetitions of one of the motor actions before passing to the each of the other ADLs) to collect data for training the model; on the other side, the execution of motor tasks lies within a circuit (i.e., ADLs are performed uninterruptedly) to record signals for testing the workflow. Moreover, the CHAR is addressed first by employing a sliding-window procedure on the pre-processed IMU signals to predict the ADL in each of these *sub-windows*, and then by combining these outcomes to obtain the final prediction related to a wider window, i.e., *grouped window*. To the best of the authors’ knowledge, this classification strategy has already been proposed in one related work, but it was not tested on a dataset of human activities performed in a continuous way [57]. Two metrics are exploited to quantitatively evaluate the performance of the proposed framework: accuracy (ACC) serves to assess the capability of the model to perform CHAR, whereas inference time (IT) is aimed at ascertaining its feasibility in a real-time monitoring scenario. As described in Section 2.2.3, all possible combinations of sensors made up of at least two IMUs were considered in order to determine the optimal sensor configuration, i.e., the number and location and sensors leading to a good compromise between classification accuracy and inference time.

The quantitative outcome related to metrics and comparisons are reported in detail in the previous subsections. Such results show that the classification strategy effectively increased accuracy for all combinations; remarkably, the exploitation of IMUs both at the right wrist and the two pelvises (e.g., RW+RP+LP combination) led to the highest boost in accuracy. This proves the efficacy of the proposed averaging strategy in enabling more accurate predictions of continuously performed human actions. Furthermore, even though employing *grouped windows* delayed the final prediction of the model targeting CHAR, the time needed to recognize one activity performed in a seamless way stayed beneath 500 ms for all combinations. Hence, inference time on a single data sample is coherent with a real-time scenario of health monitoring impaired subjects, whose execution of ADLs lasts more than 2 s [58,59,60]. In light of this, one may infer that the exploitation of *grouped windows* effectively increases the classification performance without excessively enlarging inference time for all combinations, especially the RW+RP+LP combination that is revealed to be the optimal sensor configuration as a good trade-off between accuracy and computational cost; indeed, albeit the average time needed to inference on the test dataset becomes slightly higher, such increment is not significant and the average accuracy of the model using all sensors gains from 83.83% to 96.69% after combining predictions of sub-windows.

When comparing sensor combinations (see Section 3.2 and Section 3.3), it emerged that configuration with two sensors located at the right wrist and the left pelvis, i.e., RW+LP is the best among sensor pairs, arguably because it integrates information regarding the motion of both upper and lower limbs, respectively. On the contrary, placing IMUs either at the left pelvis and the sternum (LP+S) or at the two sides of the pelvis (RP+LP) slightly decreases classification performance. This results in misclassification between S2S and LD (Figure 8), which may be due to the absence of information about the movement of the right wrist that is recruited to support the lying-down action. Indeed, integrating wrist-related information into either LP+S or RP+LP combinations (i.e., RW+LP+S and RW+RP+LP, respectively) can reduce misclassifications and ensure a more accurate CHAR-targeting framework.

On the other hand, pairing the sensor at the right pelvis with the one at the right wrist (i.e., RW+RP combination) may lead the model to fail in recognizing turning, which is confused with walking, and sit-to-stand, which is misclassified as lying-down; the sensor configuration with IMUs at the two sides of human pelvises reduces misclassifications between TN and W, keeping them between S2S and LD. On the contrary, feeding the model with data coming from both pelvises and the right wrist (i.e., RW+RP+LP triple) enhances accuracy.

In the author’s opinion, such outcomes can be explained with the following motivation: the sensor at the left pelvis captures relevant information about the turning action, whereas the one at the right wrist is crucial in recognizing the lying-down motor pattern. Hence, either using this sensor pair (i.e., RW+LP) or integrating it with one (i.e., RW+RP+LP) or more sensors (i.e., RW+RP+LP+S) allows for classifying continuously executed ADLs with a satisfying accuracy (close to 100%); however, the RW+RP+LP triple is the optimal configuration of sensors in the proposed workflow for CHAR, since it does not lead to an excessive increase in the inference time with respect to using two sensors.

## 5. Conclusions

This work presents a framework based on Deep Learning (DL) for classifying activities of daily living (ADLs) that are executed uninterruptedly by means of data coming from inertial sensors placed at different parts of the human body. Comparison of the computed metrics for different sensor configurations proved the efficacy of the proposed workflow in accurately recognizing motor actions with temporal performance that are acceptable in a real-time clinical scenario. Most notably, the outcomes indicate that the integration of sensors located at the right wrist and the two pelvises offered a good compromise between accuracy and computational cost.

The main limitation of the study is related to the size of the experimental sample. This could be addressed either by recruiting new subjects with different characteristics or by exploiting DL-based data augmentation algorithms, such as generative adversarial networks [63]. Besides, the investigation may be pushed forward by implementing explainable artificial intelligence methods (e.g., attention mechanism) to improve interpretability [64,65,66,67].

In addition, the dataset of activities to be recognized can be enlarged so as to include other clinically relevant transitional motor actions (e.g., stand-to-walk, stand-to-sit, walk-to-sit, or lying-to-sit). In so doing the proposed framework could be even addressed for evaluations of either motor or cognitive impact, such as motion intent recognition [68,69,70]. Furthermore, acquiring inertial signals during stair ambulation would offer the possibility to investigate another motor pattern in which some patients would exhibit an abnormal execution of the task due to the fear of falling [15].

## Figures and Tables

**Figure 1 sensors-24-02199-f001:**
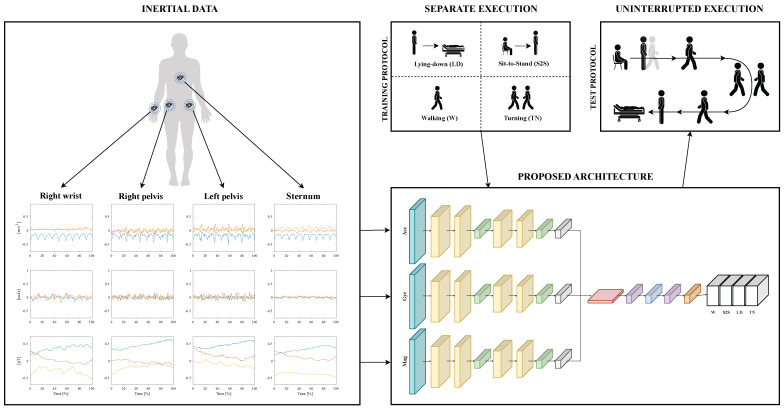
The framework oriented to CHAR: inertial data are collected from four sensors placed on the human body; each component of the IMU sensors is used to feed a multi-branch DL-based architecture; this model is trained on the data related to a separate execution of four ADLs that are Lying-down (LD), Sit-to-stand (S2S), Walking (W), and Turning (TN); then, it is tested on the signals coming from motor actions that are performed uninterruptedly.

**Figure 2 sensors-24-02199-f002:**
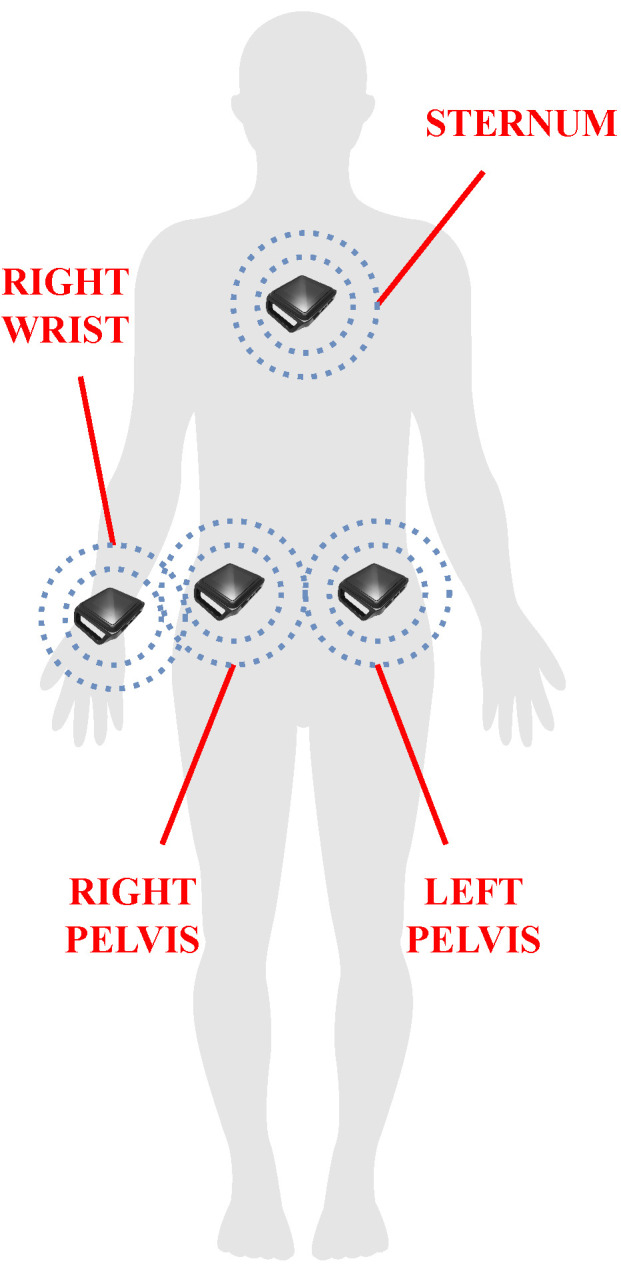
The placement chosen for the sensor in the proposed framework: two IMUs are located on the two sides of the human pelvis to monitor motor actions driving the lower limbs (e.g., walking, sit-to-stand, lying-down); one sensor on the sternum serves to register the trunk contribution to accomplishing sit-to-stand and lying-down activities; the sensor on the right wrist (i.e., the wrist of the dominant arm) aims at acquiring the possible use of hands during lying-down, as well as the arm swing while walking.

**Figure 3 sensors-24-02199-f003:**
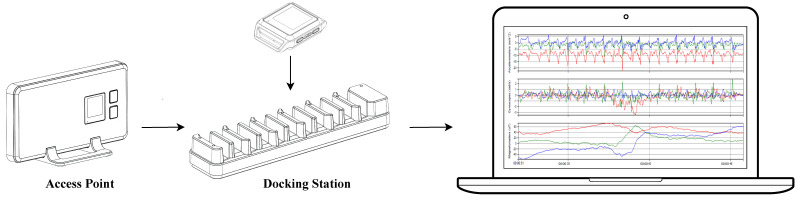
Motion Studio system with IMU sensors, Access Point, Docking Station, and PC.

**Figure 4 sensors-24-02199-f004:**
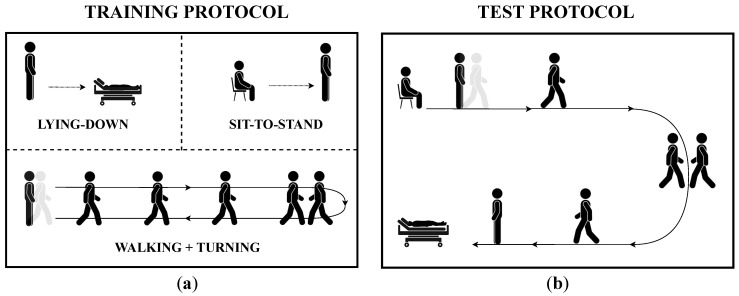
The two experimental protocols encompassed in the study with the aim of continuous HAR: (**a**) the protocol for acquiring the training dataset encompasses an interrupted execution of motor tasks, which are Lying-down, Sit-to-stand, and Walking+Turning; (**b**) the protocol for collecting the test dataset requires subject to perform the same tasks without interruptions on a predefined path.

**Figure 5 sensors-24-02199-f005:**
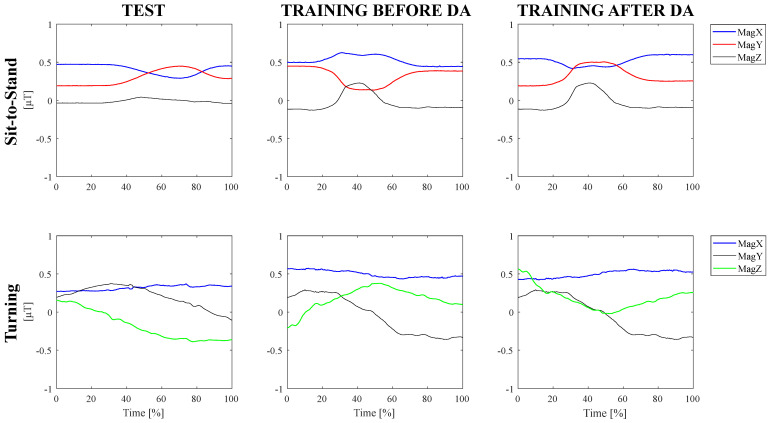
Data augmentation technique to cope with operational differences.

**Figure 6 sensors-24-02199-f006:**
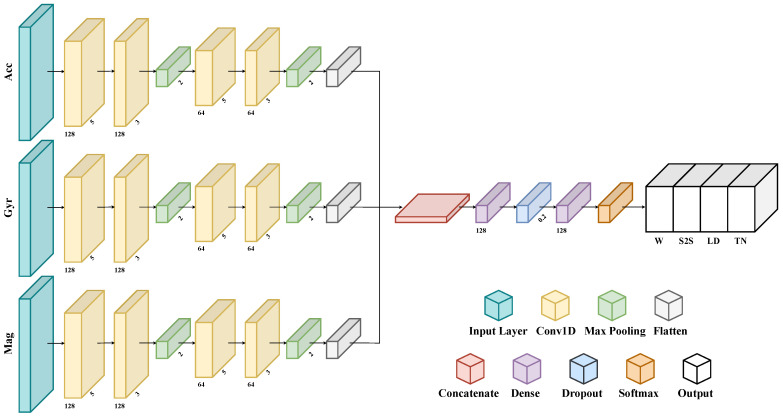
The custom multi-branch CNN addressing CHAR.

**Figure 7 sensors-24-02199-f007:**
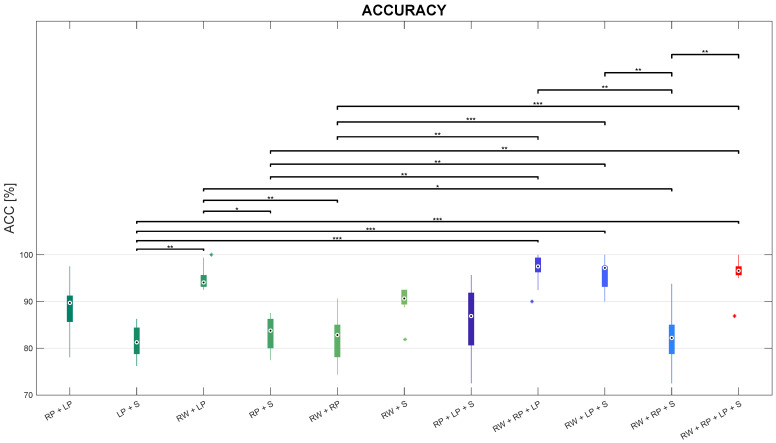
Boxplots of accuracy distributions on the test set for each sensor combination, with *, **, and *** representing statistically significant comparisons with p<0.05, p<0.01, and p<0.001, respectively.

**Figure 8 sensors-24-02199-f008:**
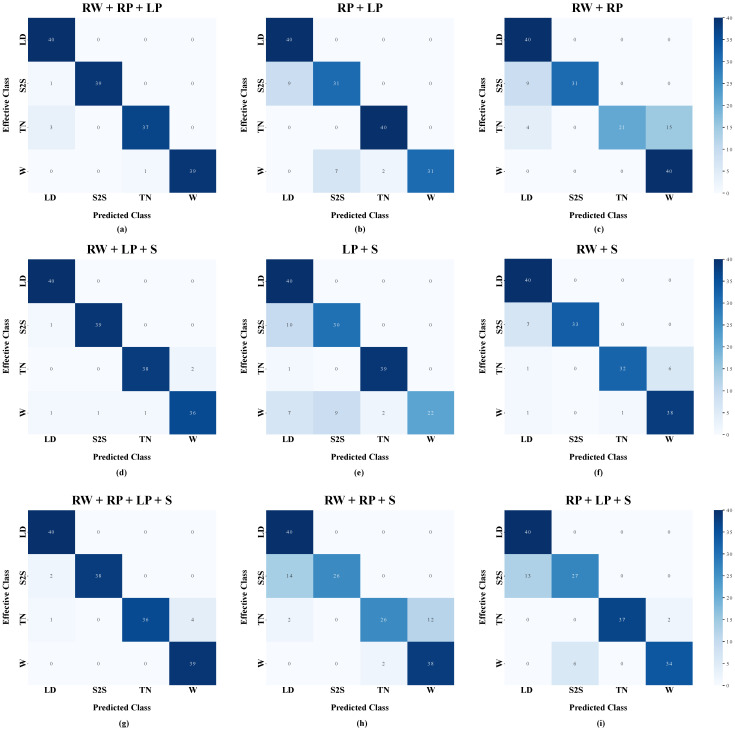
Confusion matrices of the proposed model for continuous human activity recognition for three sensor configurations: (**a**) RW+LP+S means that inertial sensors are placed at the right wrist, the left pelvis, and the sternum; (**b**) LP+S means that inertial sensors are placed at the left pelvis and the sternum; (**c**) RW+S means that inertial sensors are placed at the right wrist and the sternum; (**d**) RW+RP+LP means that inertial sensors are placed at the right wrist, and the right and left pelvises; (**e**) RP+LP means that inertial sensors are placed at the right pelvis and left pelvis; (**f**) RW+RP means that inertial sensors are placed at the right wrist and the right pelvis; (**g**) RW+RP+LP+S means that inertial sensors are placed at the right wrist, the right and left pelvises, and the sternum; (**h**) RW+RP+S means that inertial sensors are placed at the right wrist, the right pelvis, and the sternum; (**i**) RP+LP+S means that inertial sensors are placed at the right and left pelvises and the sternum. The activities to be recognized are Lying-down (LD), Sit-to-stand (S2S), Turning (TN), and Walking (W).

**Figure 9 sensors-24-02199-f009:**
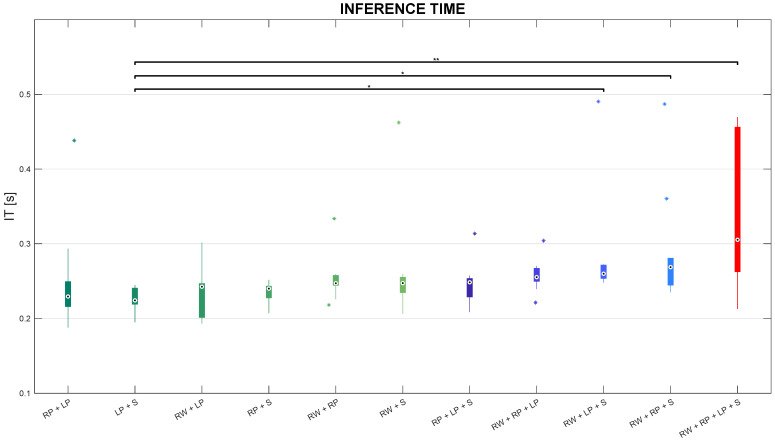
Boxplots of inference time distributions on the test set for each sensor combination, with * and ** representing statistically significant comparisons with p<0.05 and p<0.01, respectively.

**Table 1 sensors-24-02199-t001:** Average accuracy [%] on ten-fold testing sets for each combination of sensors with the two window types (e.g., *sub-windows* and grouped windows). The corresponding *p*-values are specified with a significance level set to p<0.05 as well.

Combination	Test Accuracy [%]		*p* Value
	Sub-Windows	Grouped Window	
RP+LP	75.42	88.75	0.0001
LP+S	65.94	81.44	0.0001
RW+LP	79.59	95.00	0.0001
RP+S	78.01	83.31	0.0298
RW+RP	79.64	82.31	0.1569
RW+S	77.74	89.88	0.0001
RP+LP+S	74.90	86.12	0.0018
RW+RP+LP	83.83	96.69	0.0001
RW+LP+S	80.33	95.75	0.0001
RW+RP+S	79.16	81.38	0.4212
RW+RP+LP+S	84.88	96.06	0.0001

**Table 2 sensors-24-02199-t002:** Average and standard deviation inference time in seconds needed for the inference related to ten-fold testing sets of each window type with the two window types (e.g., sub-windows and grouped window). The corresponding *p* Values are specified with significance level set to p<0.05 as well.

Combination	Inference Time [s]		*p* Value
	Sub-Windows	Grouped Window	
RP+LP	0.21±0.04	0.25±0.07	0.1901
LP+S	0.22±0.02	0.22±0.02	0.3565
RW+LP	0.22±0.02	0.24±0.03	0.1820
RP+S	0.23±0.01	0.25±0.03	0.0684
RW+RP	0.23±0.07	0.25±0.03	0.4002
RW+S	0.23±0.03	0.26±0.07	0.1660
RP+LP+S	0.23±0.03	0.25±0.03	0.1678
RW+RP+LP	0.24±0.07	0.26±0.02	0.4601
RW+LP+S	0.24±0.03	0.28±0.07	0.1206
RW+RP+S	0.26±0.05	0.29±0.08	0.2706
RW+RP+LP+S	0.30±0.09	0.33±0.10	0.3803

## Data Availability

The data presented in this study are available on request from the corresponding author.

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
