# Peer review of "A Novel Framework Based on Deep Learning Architecture for Continuous Human Activity Recognition with Inertial Sensors"

_sensors, 2024, doi:10.3390/s24072199_

Round 1

Reviewer 1 Report

Comments and Suggestions for Authors

Nice paper and well presented. In the main the manuscript has some merit but would be improved from addressing the following points and shortened. There are some major limitations.

1. "Subjects with different physical characteristics are recruited to increase variability in age and anthropometric characteristics" that is rather vague. What do you mean by physical characteristics? I presume all fit and healthy (i.e., no functional impairments) so that suggests no major differences to reflect arising data heterogenity.

2. Better to describe that the wrist was place on the dominent arm. 

3. Are the logo's in Fugure 3 used with permission from APDM?

4. Training dataset - as per #1 above, what is the heterogentiy of the people used for training data to help generalizability of the arising model? Equally, 10 is a very small number and the number of physical tasks performed very few... this is a major limittaion of the paper. For contemporary research in this field I would expect to see many people (e.g. >50) recruited and IF wanting to use clinically, across different patient groups (but appreciate the clinic aspect might be future work).

5. What was the rationale for placing a device on the chest. From a technical perspective it makes HAR a bit easier but in reality, people do not want to stick/strap an IMU at that location - so why not consider the leg (would help with e.g., lying recognition)? the rationale for IMU place is not fully clear.

6. Why doesn't the protocol have stair navigation? that seems to a major flaw as HAR for stair ambulation in rehabilitation is key when trying to understand e.g., falls.

Also - the volume of data is VERY small, data collection was about 1min/person? That is much to short in comparison to other studies who would also test beyond controlled settings to included scripted events but in ecologically valid situations. 

But data were also augmented to increased the base volume?

7. The confusion matrices are too "small" would suggest 3x1 rather than 1x3 layout (and increased along legth of page rather than width).

8. the paper has merit but is more suited to a "technical note" rather than the lengthy version in which it is currently presented.

Reviewer 2 Report

Comments and Suggestions for Authors

This paper proposes a novel framework based on deep learning architecture for continuous human activity recognition with inertial sensors. The framework performs continuous human activity recognition by training a custom 1D-CNN with IMU signals that are related to a separate execution of activities of daily living, and testing it on inertial data recorded during an uninterrupted execution of those activities. Further, a classification strategy is proposed to enhance accuracy without increasing significantly the time needed recognize activity. However, some parts in the paper would need to be carefully revised.

1. A total of 10 male subjects are recruited in line 162, which is inconsistent with the number of male subjects in lines 228 and 248.

2. The experimental protocol for the test dataset is made up of the same ADLs as those of the training dataset. The subject turns 180 degrees counterclockwise around a pin in the Walking+Turning task of training set. However, why is it clockwise in the test set? Please explain.

3. In lines 302-335, the magnetometer signal is flipped. Then how to do data augmentation processing for accelerometer signal and gyroscope signal? Please explain.

4. There are some typos in the paper. For example, Figure 8, “LD”might be “LyD”. In Figure 9, the meaning of "RP+LP" might be incorrectly described, “right wrist” might be “right pelvis”.

Comments on the Quality of English Language

Minor editing of English language required.

Round 2

Reviewer 1 Report

Comments and Suggestions for Authors

The authors have replied and performed a few changes but they are quite minimal. To me, the revised work doesn't go far enough to improve the work to a leading standard. The work is still "ok" only - the authors haven't really taken on board the feedback.